# Emergent Ashkin-Teller criticality in a constrained boson model

Anirudha Menon[1], Anwesha Chattopadhyay[2], K. Sengupta[1], and Arnab Sen[1]

1 School of Physical Sciences, Indian Association for the Cultivation of Science, Kolkata 700032, India
2 Department of Physics, School of Mathematical Sciences, Ramakrishna Mission Vivekananda Educational and Research Institute, Belur, Howrah 711202, India.
* ksengupta1@gmail.com

November 5, 2024

## Abstract

We show, via explicit computation on a constrained bosonic model, that the presence of subsystem symmetries can lead to a quantum phase transition (QPT) where the critical point exhibits an emergent enhanced symmetry. Such a transition separates a unique gapped ground state from a gapless one; the latter phase exhibits a broken $Z_2$ symmetry which we tie to the presence of the subsystem symmetries in the model. The intermediate critical point separating these phases exhibits an additional emergent $Z_2$ symmetry which we identify; this emergence leads to a critical theory in the Ashkin-Teller, instead of the expected Ising, universality class. We show that the transitions of the model reproduces the Askhin-Teller critical line with variable correlation length exponent $\nu$ but constant central charge $c$. We verify this scenario via explicit exact-diagonalization computations, provide an effective Landau-Ginzburg theory for such a transition, and discuss the connection of our model to the PXP model describing Rydberg atom arrays

# 1 Introduction

Symmetries play a pivotal role in shaping the properties of phases and phase transitions in generic many-body systems [1]. These symmetries can be global or local; the former can be spontaneously broken leading to typical symmetry-broken phases of matter such as magnets or superconductors. The local or gauge symmetries, in contrast, are always preserved [2]; however they can lead to local constraints shaping properties of low-energy phases. A well-known example of such a phenomenon involves spin-ice systems [3].

More recently, a class of symmetries which are intermediate between these two types have been identified. They manifest on $(d > 0)$-dimensional subsystems of a $(D > d)$-dimensional system and are, therefore, called subsystem symmetries. In this method of classification, the gauge (global) symmetries corresponds to $d = 0(D)$. Such subsystem symmetries play a key role in field theories which feature dimensional reduction and in physics of fractons [4–6]. It has been shown that such symmetries can be spontaneously broken [7,8].

The above-mentioned features elicit a natural question regarding role of subsystem symmetries in shaping properties of quantum phases and transitions between them. This question has been partially addressed in Ref. [9], where a quantum critical point with fractal symmetry was identified in the Newman-Moore model [10]. In addition, disordered Ising system with subsystem symmetries are shown to have first order transitions whose properties are shaped by the presence of such symmetries [11,12].

In this work, we demonstrate, using a constrained boson model, that subsystem symmetries may change the universality class of a quantum phase transition (QPT). Our model hosts a gapped and a gapless phases separated by a QCP (Fig. 1). The gapless phase has a $Z_2$ symmetry-broken ground state which we show to be a consequence of the subsystem symmetry of the model. Remarkably, the intermediate QCP has an enhanced $Z_2 \times Z_2$ symmetry and belongs to the Ashkin-Teller (AT), rather than the expected Ising, universality class [13–24]. The additional $Z_2$ symmetry at the QCP is emergent; we identify the root of this symmetry using an effective Landau-Ginzburg theory and discuss its relation with the subsystem symmetries of the model. We explicitly compute, using exact-diagonalization (ED) on finite quasi one-dimensional (1D) arrays, the dynamical critical exponent $z$, correlation length exponent $\nu$, and the entanglement entropy $S_E$ and determine the central charge $c$ of the conformal field theory (CFT) governing the QCP. Our analysis identifies a parameter $\alpha$ of the model which can be varied to access the AT critical line with variable $\nu(\alpha)$ but constant $c$ [22,25]. We also discuss the relation of the boson model with the well-known PXP model describing experimentally realizable Rydberg atom arrays [26–32]. Our model leads to realization of AT criticality in a 1D model with Hilbert space dimension (HSD) $\zeta^L$ for $L$ sites, where $\zeta = (1+\sqrt{5})/2$ is the golden ratio. This is much smaller than HSD for other realizations of the AT model [13–24]. We therefore expect it to provide significant numerical advantage in simulating physics of AT criticality.

# 2 The model

We consider a model of hardcore bosons on an array of $N_p$ square plaquettes (Fig. 1). These bosons have an additional constraint; any neighboring sites $j = (j_x, j_y)$ and $j + n = (j_x \pm 1, j_y)$ or $(j_x, j_y \pm 1)$ can not be simultaneously occupied. The longitudinal (transverse) diagonals of these plaquettes are designated by $d_l$ ($d_t$). We also define the two-boson states

$$|\ell_j\rangle = b_j^\dagger b_{j+d_l}^\dagger |0_j; 0_{j+d_l}\rangle, \quad |t_j\rangle = b_j^\dagger b_{j+d_t}^\dagger |0_j; 0_{j+d_t}\rangle \tag{1}$$

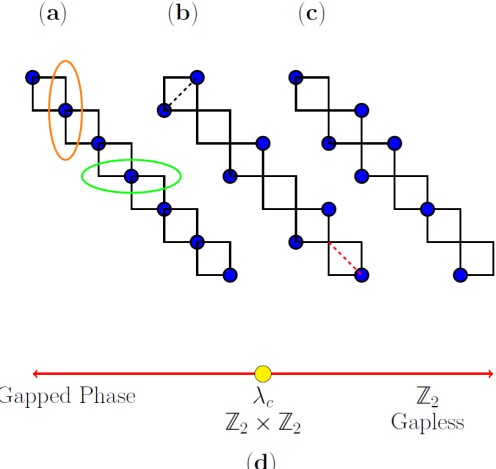

Figure 1: Schematic representation of classical Fock states of the bosons model. The longitudinal(transverse) $[d_\ell(d_t)]$ diagonals are shown by dashed red (black) lines. A typical set of horizontal(vertical) connected links are shown by green(orange) ellipses. (a) Ground state at $\lambda/w \to -\infty$ for $N_p = 2n = 6$ plaquettes and total boson number $N = N_p + 1$. (b) One of the classical states forming the ground state manifold for $\lambda/w \to \infty$. (c) A state with two bosons on a vertical and a horizontal link; these are dynamically disconnected from those shown in (a) and (b). (d) Schematic phase diagram of the model as a function of $\lambda/w$ showing gapped and gapless $Z_2$ symmetry broken phases separated by a quantum critical point (yellow circle) with enhanced $Z_2 \times Z_2$ symmetry at $\lambda = \lambda_c$. See text for details.

Here $b_j^\dagger$ denotes the boson creation operator on site $j$ of the plaquettes, $|..0_j..\rangle$ indicates a Fock state having zero bosons on site $j$, and we define the boson number operator to be $\hat{n}_j = b_j^\dagger b_j$. The constraint condition necessitates $\hat{n}_j \hat{n}_{j+n} = 0$ for all $j$. The boson Hamiltonian can then be written as $H = H_0 + H_1$ where

$$
\begin{aligned}
H_0 &= -w \sum_j (|t_j\rangle\langle \ell_j| + \text{h.c.}) \\
H_1 &= \lambda \sum_j (|\ell_j\rangle\langle \ell_j| - \alpha |t_j\rangle\langle t_j|),
\end{aligned}
\tag{2}
$$

where $w, \alpha > 0$ and $\lambda$ can have either sign. The variation of the parameter $\alpha$, as shall be elucidated later, allows one to access the AT critical line. We note that $H_0$ denotes a ring-exchange term for the bosons; it's implementation on a 2D square lattice is known to show strong Hilbert space fragmentation [33]. $H_1$ has a different sign for bosons occupying $d_\ell$ or $d_t$; this feature will be important for our purpose. In this work, we seek the phase diagram of this model as a function of $\lambda/w$ and $\alpha$. In what follows, we shall consider an array (Fig. 1) with open (periodic) boundary condition having even $N_p$ plaquettes hosting $N = \sum_j n_j = N_p + (-)1$ bosons.

The model given by Eq. 2 has $[H, N] = 0$ leading to preservation of the total boson number. For open chains, the eigenstates of $H$, $|n_\pm\rangle$, can be labelled by their parity eigenvalues $r = \pm 1$ such that $R|n_\pm\rangle = \pm 1|n_\pm\rangle$. Here $R$ represents reflections about an axis along $d_t$ dividing the system in two equal halves. Importantly, $R$ is the *only* global symmetry that can be spontaneously broken for open chains; thus, it can lead to a $Z_2$ broken phase. For periodic boundary conditions, the eigenstates can be labeled by momentum $k$ along $d_\ell$.

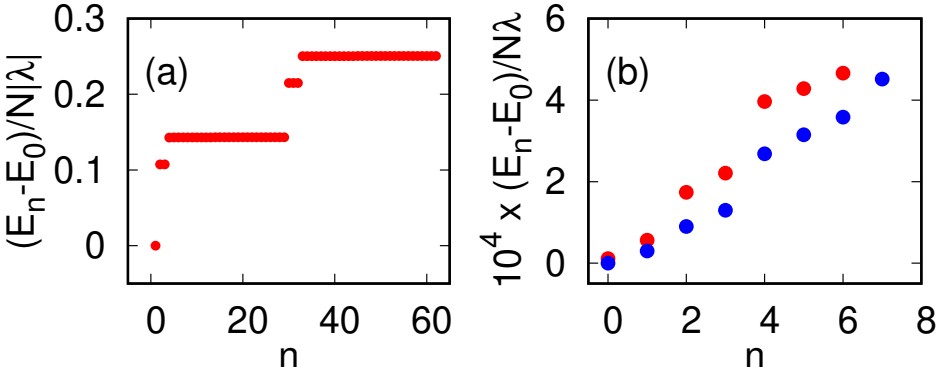

Figure 2: (a) Plot of eigenenergies of $H$ for (a) $\lambda/w = -10$ showing the end states and the first excited state manifold and (b) $\lambda/w = 10$ showing dispersing band of eigenstates. The ground state for (b) is two-fold degenerate and gapless in thermodynamic limit. For all plots $N - 1 = N_p = 28$ and $\alpha = 1$. See text for details.

Moreover, the model has subsystem symmetry which preserves $n_0 = \sum_{j \in q} n_j$, where $q$ denotes any vertical or horizontal link with three sites as shown in Fig. 1(a)(except the links involving the end sites of an open chain that contain only two sites) for all links. In what follows we shall work in the sector $n_0 = 1$ for all such links. We note that for $N = N_p \pm 1$, $n_0 = 1$ is the only possible subsystem symmetry sector of the model.

## 3 Phases of the model

In this section, we discuss the phase of $H$ (Eq. 2) in the two limit $\lambda < 0$, $|\lambda| \gg w$ and $\lambda \gg w$ using ED in Sec. 3.1. This is followed by a perturbation theory in Sec. 3.2 which provides analytic insight into the nature of the phase for $\lambda \gg w$.

### 3.1 Numerical results

For $\lambda < 0$ and $|\lambda| \gg w$, the ground state of $H$ constitutes all bosons arranged along $d_\ell$ as schematically shown in Fig. 1(a). The first excited state can be created by action of $H_0$ on one of the end sites with energy cost $\Delta E_1 = |\lambda|(2 + \alpha)$; this leads to two end states (where the plaquette with bosons on transverse diagonals is at one of the ends) shown in Fig. 2(a) for $\alpha = 1$. The action of $H_0$ on any of the bulk sites lead to states with $\Delta E_2 = |\lambda|(3 + \alpha) > \Delta E_1$. Thus we have an unique gapped ground state. The end states are protected from the rest of the bulk states by a finite energy gap and are therefore stable against small perturbations. The ground state entanglement entropy, $S_E$, shows the expected area-law behavior.

In contrast, for $\lambda \gg w$, the classical ground state manifold is $N_p/2$ fold degenerate and comprises of Fock states with maximal bosons across $d_t$ (Fig. 1(b)). The conservation of $n_0$ ensures that two such plaquettes can not be neighbors; such states belong to a sector with $n_0 = 2$ on at least one vertical and horizontal links (Fig. 1(c)) and are dynamically disconnected from the $n_0 = 1$ sector. These Fock states in the $n_0 = 1$ sector can be characterized by the position of the single bosons $|j_0\rangle$ (Fig. 1(b)). We note that $j_0$ is either always even or always odd and can take $N_p/2$ values; this leads to a $N_p/2$-fold degenerate classical ground state manifold. This classical degeneracy is lifted by quantum fluctuations induced by $H_0$. The effect of $H_0$ can be understood within perturbation theory which we discuss in the next subsection. Here we note that the spectrum of the model, obtained using ED for a chain with $N_P = 28$ plaquettes and

open boundary conditions, shows two near degenerate bands (Fig. 2(b)). For open boundary condition, the presence of the subsystem symmetry ensures a two-fold reflection symmetry $R$ which protects this degeneracy with small energy splitting between $R = \pm 1$ states for finite $N_p$ (Fig. 2(b)) that decrease with $N_p$. Thus we find a $Z_2$ symmetry-broken gapless ground state for $\lambda \gg w$. We have checked that $S_E$ shows logarithmic dependence on the subsystem size $\ell_p$ in this phase.

### 3.2 Perturbation theory

In this section, we compute an effective Hamiltonian, $H_{\text{eff}}$, for $\lambda \gg w$ using perturbation theory which explains the properties of the ground state manifold for $\lambda \gg w$. To this end, we note that in the limit $w \to 0$ for any $\lambda/w > 0$, the ground state has $N_p/2$-fold degeneracy. To lift this degeneracy, one needs to take into account the effect of $H_0$. To this end, we seek a canonical transformation via an operator $S$ which satisfies [33]

$$H_{\text{eff}} = e^{iS}(H_1 + H_0)e^{-iS}, \tag{3}$$

such that $H_{\text{eff}}$ does not take one outside the low-energy manifold. Usually it is impossible to find an exact solution for $S$ and hence $H_{\text{eff}}$; however, this can be achieved perturbatively and we shall chart out such a perturbative calculation where $w/\lambda$ is the small parameter. In this regime, we define $S$ such that all first order ($O(w/\lambda)$) processes that take one outside the low-energy manifold of states $|j_0\rangle$ of $H_1$ are eliminated. This leads to the solution [33]

$$[iS, H_1] = -H_0, \quad H_{\text{eff}} = \frac{1}{2}[iS, H_0], \tag{4}$$

where we have omitted all processes which are $O(w^4/\lambda^4)$ or higher. To proceed further, we note that from the first relation in Eq. 4, we find that the matrix elements of $S$ between any state $|j_0\rangle$ in the ground state manifold with energy $E_0$ and an excited state $|\beta\rangle$ outside this manifold with energy $E_\beta > E_0$ is given by

$$\langle \beta | iS | j_0 \rangle = \frac{\langle \beta | H_0 | j_0 \rangle}{E_\beta - E_0} \tag{5}$$

Note that $H_0$ and hence $S$ do not connect states within the low-energy manifold. Using Eq. 5 we find

$$H_{\text{eff}} = -\sum_{j_0 j_0', \beta \neq j_0, j_0'} \frac{\langle j_0 | H_0 | \beta \rangle \langle \beta | H_0 | j_0' \rangle}{E_\beta - E_0} |j_0\rangle \langle j_0'|. \tag{6}$$

This yields the leading order terms in the effective Hamiltonian.

The evaluation of the matrix elements in Eq. 6 is straightforward. As explained in Fig. 3, there are two processes in the leading order. The first, schematically shown in Fig. 3(a) leads to renormalization of the energy of eigenstates in the ground state manifold and do not contribute towards lifting the degeneracy. The process which lifts this degeneracy is schematically shown in Fig. 3(b); the energy of the high-energy intermediate state for this process is $(2 + \alpha)\lambda$ leading to a matrix element of $w^2/((2+\alpha)\lambda)$. Furthermore such processes constitutes hopping of the lone boson by two lattice sites and thus connects states $|j_0\rangle$ to $|j_0'\rangle = |j_0 \pm 2\rangle$. These considerations leads to he effective Hamiltonian

$$H_{\text{eff}} = -\frac{w^2}{(2+\alpha)\lambda} \sum_{j_0} (|j_0\rangle \langle j_0 - 2| + \text{h.c.}), \tag{7}$$

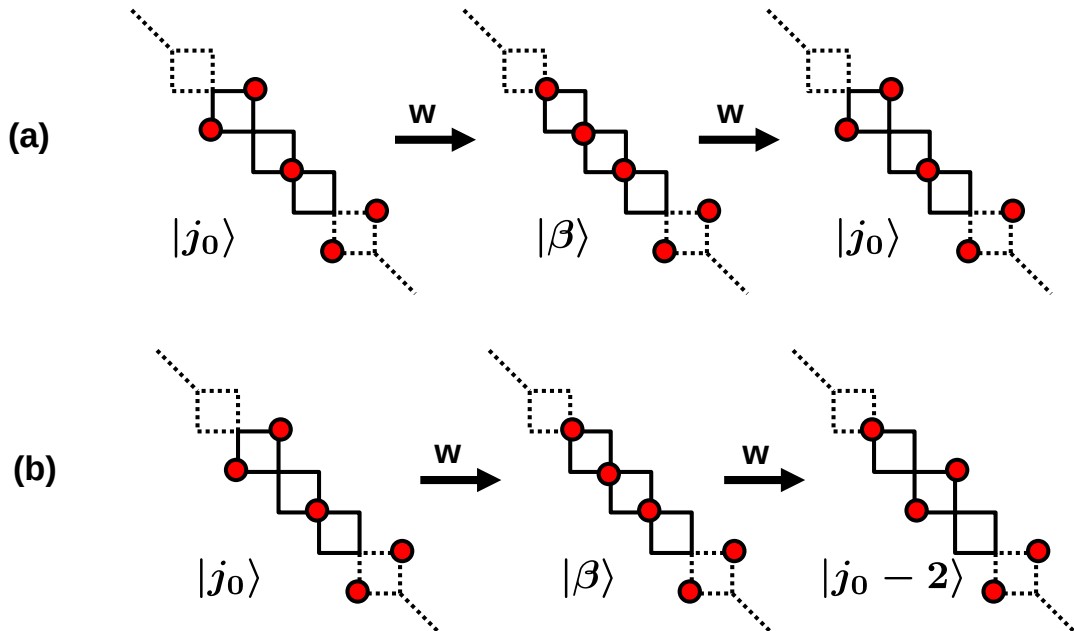

Figure 3: (a) Schematic representation of a second-order virtual process connecting a state $|j_0\rangle$ in the ground state manifold to itself leading to renormalization of their energy. (b) Same for a process which connects $|j_0\rangle$ to $|j_0 - 2\rangle$ and thus contributes to $H_{\text{eff}}$. In both plots, $|\beta\rangle$ denotes a state outside the ground state manifold of $H_1$ with $E_\beta - E_0 = (2 + \alpha)\lambda$. The dashed blue plaquettes do not participate in the virtual process and the bosons are denoted by red circles. See text for details.

The leading term in $H_{\text{eff}}$ connects classical states where the position of the single boson is two lattice sites apart. To analyze this further, we note that chains with periodic boundary conditions and $N = N_p - 1$ shows identical behavior to open chains with $N = N_p + 1$. For the former, $H_{\text{eff}}$ lead to energy dispersion

$$E(k) = -\frac{2w^2}{(2 + \alpha)\lambda} \cos 2k + \mathrm{O}(w^4/\lambda^3), \tag{8}$$

where we have set the length of the diagonals of the plaquette to unity. Since $E(k) = E(\pi - k)$, one has a two-fold degeneracy which can be lifted only by terms of $\mathrm{O}(w^{N_p}/\lambda^{N_p - 1})$. This degeneracy stems from the fact that in the $n_0 = 1$ sector, the dynamically connected states always have $j_0$ either even or odd; it is a consequence of subsystem symmetry of $H$.

We note that finite higher order processes do not lift the above-mentioned degeneracy. For a chain with periodic boundary condition, this is most easily seen from the fact that there are no odd-order ($\mathrm{O}(w^{2n+1}/\lambda^{2n})$) terms in the perturbation series which, starting from an initial state in the ground state manifold, can return to the same manifold. Thus all processes which leads to finite matrix elements between states in the ground state manifold are even-ordered ($\mathrm{O}(w^{2n}/\lambda^{2n-1})$). Next, we find that any such even-ordered term necessarily leads to shift in the position of the lone bosons by a multiple of two-sites. This follows from the fact that the lowest such term leads to a shift of this boson by two sites; one such higher order process which leads to a shift of the boson by four sites is shown in Fig. 4. Such a process connects $|j_0\rangle$ to $|j_0 \pm 4\rangle$. Similar considerations hold for all high order virtual processes; this ensures that the two-fold degeneracy is not lifted at any order by $H$.

Next we discuss the degeneracy in case of a boson array with open boundary condition. This degeneracy arises to the inversion symmetry $R$ which, for a chain with even $N_p$, constitutes a reflection about a line along the transverse diagonal $d_t$ that cuts the array into two

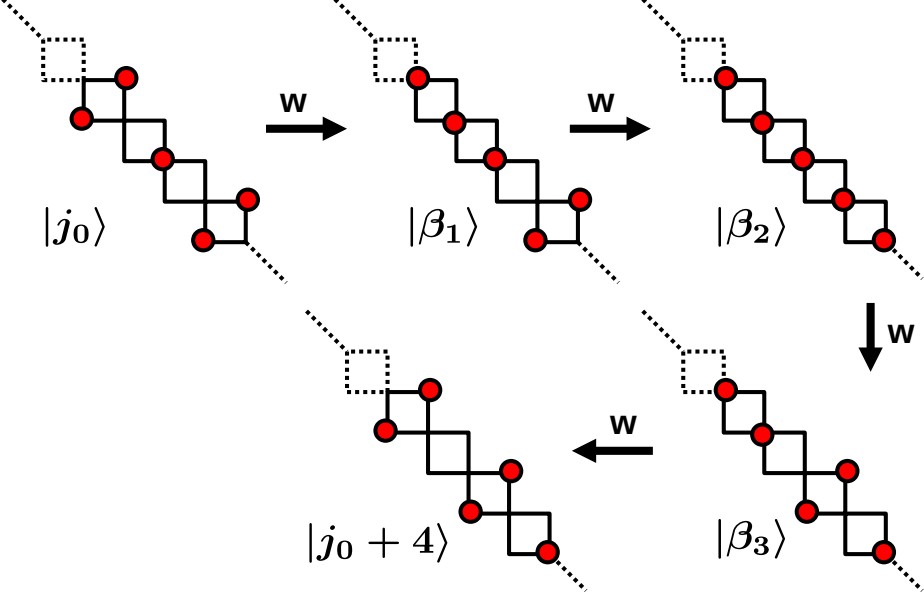

Figure 4: Schematic representation of a fourth-order virtual process connecting a state $|j_0\rangle$ in the ground state manifold to $|j_0 + 4\rangle$. Here $|\beta_1\rangle$, $|\beta_2\rangle$ and $|\beta_3\rangle$ represent excited states of $H_1$ with energies $(2 + \alpha)\lambda$, $(4 + 2\alpha)\lambda$, $(1 + \alpha)\lambda$ respectively with respect to the ground state manifold. All symbols are same as in Fig. 3. See text for details.

equal halves. It is easy to see that $H$ remains invariant under such a transformation [33]. We note that the ground state at large negative $\lambda$ is an exact eigenstate of $R$ with eigenvalue $r = 1$; however, this is not the case with classical states occupying the ground state manifold at large positive $\lambda$. Numerically, for $\lambda/w \gg 1$, we find that the ground state corresponds to a linear combination of the classical Fock state with $r = 1$. The energy gap between this state and its $r = -1$ counterpart vanishes in the thermodynamic limit leading to a two-fold degeneracy. However, for a finite size array of length $N_p$, a $O(N_p)$ virtual process originating from $H_0$ connects these states leading to a gap as can be seen from Fig. 2(c). This gap decreases algebraically with system size for the range of $N_p$ we have numerically studied and also decreases with decreasing $w/\lambda$. We note that this behavior is in sharp contrast with that of the energy gap between the ground and the first excited state at large negative $\lambda$; the latter remains stable in the thermodynamic limit and does not show appreciable change with increasing $N_p$.

## 4   Quantum critical point

The ground state of the boson model for $\lambda \gg w$ breaks a $Z_2$ symmetry. In contrast, that for $\lambda < 0$ and $|\lambda| \gg w$, does not break any symmetry. Thus it is excepted that these states will be separated by a phase transition. In this section, we investigate this transition in details. Our numerics on finite chain with $N_p \leq 28$, in Sec. 4.1, demonstrates that the transition, in contrast to the expected Ising, belongs to the AT universality class; the critical point hosts an additional emergent $Z_2$ symmetry. In Sec. 4.2, we provide a Landau-Ginzburg theory for the transition which provides an analytic understanding of the mechanism for the additional emergent symmetry.

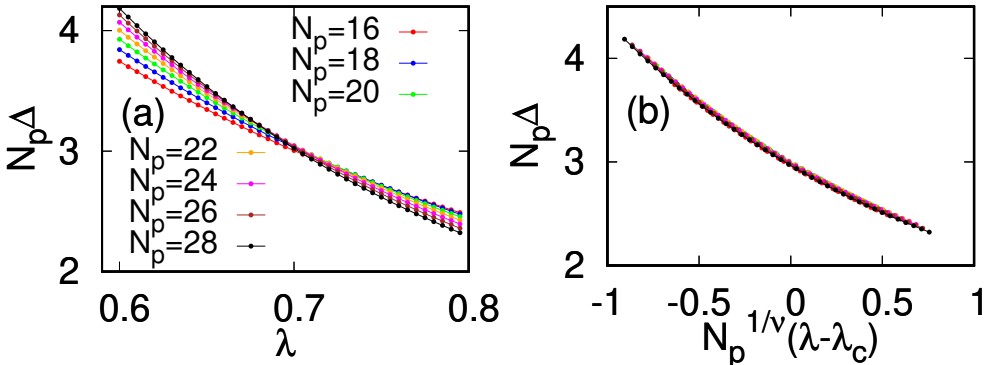

Figure 5: (a) Plot of $\Delta N_p$ as a function of $\lambda$ obtained using ED; the curves cross at $\lambda = \lambda_c \simeq 0.7$ indicating $z = 1$. (b) Plot of $\Delta N_p$ as a function of $N^{-1/\nu}(\lambda - \lambda_c)$ leading to optimal scaling collapse for $\nu \simeq 1.53211 \pm 0.01246$ and $\lambda_c/w \simeq 0.69$. For all plots $N - 1 = N_p = 28$ and $\alpha = 1$. See text for details.

## 4.1 Numerical results

To investigate the nature of this QCP between the $Z_2$ symmetry broken ground state at $\lambda > 0$ and the unique ground state at $\lambda < 0$, we carry out standard finite-size scaling analysis using ED for the boson chain with open boundary condition. The results obtained from such a study is shown in Figs. 5 and 7. It is well-known that the smallest energy gap $\Delta \equiv \Delta_{10} = (E_1 - E_0)$ ($E_{n-1}$ denotes the $n^{\text{th}}$ energy eigenvalues) for a finite chain near the critical point satisfies

$$\Delta = N_p^{-z} f(N_p^{1/\nu}(\lambda - \lambda_c)), \tag{9}$$

where $f$ denotes a scaling function, $z$ is the dynamical critical exponent, $\nu$ is the correlation length exponent and $\lambda_c$ is the critical value of $\lambda$. Our results, shown in Fig. 5 for a representative value $\alpha = 1$, indicate an excellent match with the above form of $\Delta$ for $z = 1$ and $\nu = 1.53211 \pm 0.01246$. Similar matches were found for other values of $\alpha$; the $\alpha \to \infty$ limit,as we shall see in the next section, constitutes a realization of the PXP model and will be separately discussed in Sec. 5. The value of $\nu$ found in Fig. 5 is clearly inconsistent with expected Ising universality class; in contrast, it is consistent with AT universality.

To further confirm that this transition is distinct from its Ising counterpart, we first note that a QCP which belongs to AT universality with $Z_2 \times Z_2$ symmetry is expected to have three gapless modes at $\lambda_c$; consequently, $\Delta_{20} N_p$ and $\Delta_{30} N_p$ is also expected to show finite-size scaling behavior similar to $\Delta$. This distinguishes such a critical point from one with Ising symmetry. Thus a signature of the AT universality may be found by studying scaling behavior of the higher energy gaps near the critical point. To this end, we define the gap $\Delta_{n0} = E_n - E_0$ and plot $\Delta_{20} N_P$ and $\Delta_{30} N_p$ (where $N_p$ denotes the number of plaquettes) as a function of $\lambda$ (Figs. 6(a) and (b)). These plots indicate the presence of a crossing although the position of the crossing point shifts towards larger value of $\lambda$. This can be attributed to finite-size effect since the scaling behavior of these higher gaps is expected to have stronger system size dependence leading to a strong finite-size shift of the crossing points towards larger $\lambda$. In contrast, similar results for Ising criticality, obtained by performing ED on an Ising chain with

$$H_{\text{Ising}} = -J \sum_{\langle ij \rangle} \sigma_i^z \sigma_j^z - B \sum_j \sigma_j^x \tag{10}$$

does not show any crossing for $\Delta_{20}$; only $\Delta_{10}$ shows this feature at $J/B = 1$ (Figs. 6(c) and (d)). The behavior of the latter plots (Figs. 6(c) and (d)) is qualitatively distinct from the

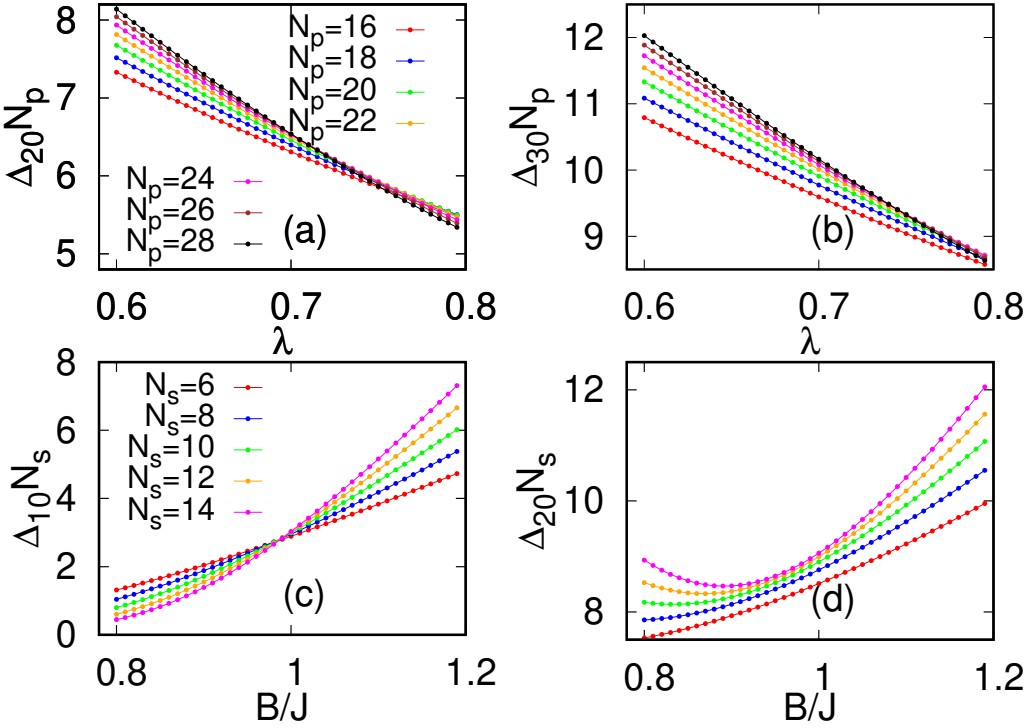

Figure 6: Plot of (a) $\Delta_{20}N_p$ (a) and (b) $\Delta_{30}N_p$ as a function of $\lambda$ showing crossings at $\lambda_c/w = 0.70925 \pm 0.00636$ and $0.74544 \pm 0.0273375$ respectively. (c) and (d) In contrast, Ising model shows such crossing only for $\Delta_{10}N_p$ as a function of $J/B$ but not for $\Delta_{20}N_p$. See text for details.

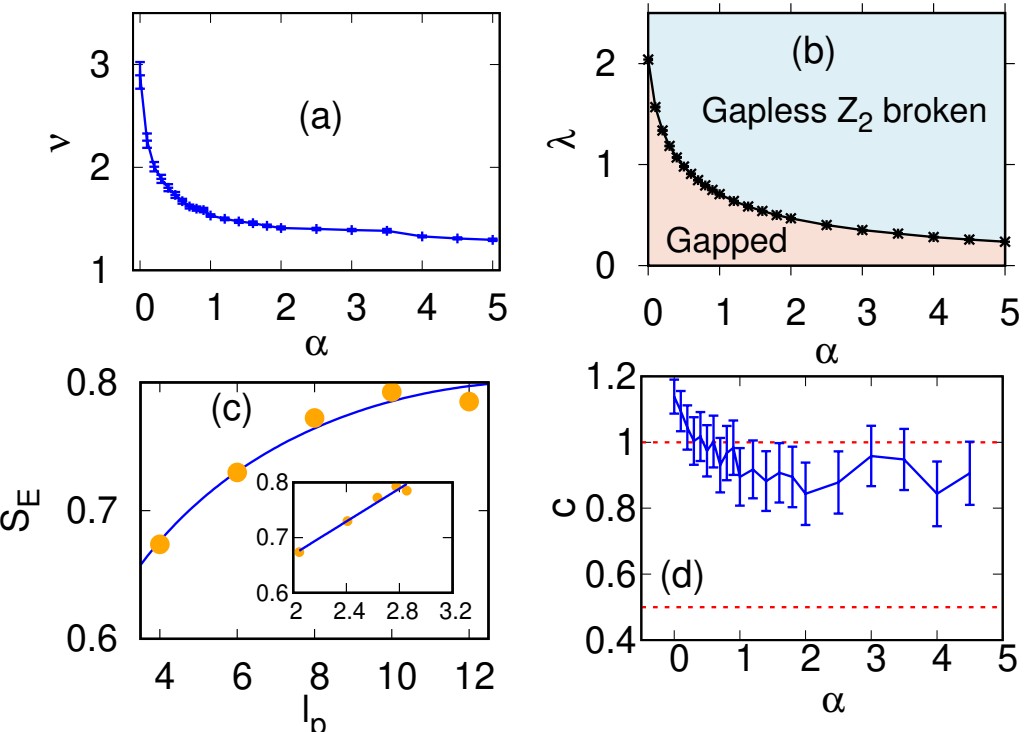

Figure 7: (a) Plot of $\nu$ as a function of $\alpha$. (b) Phase diagram in $\lambda - \alpha$ plane showing the AT critical line. (c) Plot of $S_E$ as a function of $l_p$ showing logarithmic dependence of $S_E$ on $l_p$; the inset show a plot of $S_E$ as a function of $\ln[(2N_p/\pi)\sin(\pi l_p/N_p)]$ leading to $c = 0.895072 \pm 0.08725$. (d) Plot of $c$ as a function of $\alpha$. The red dotted lines indicate $c = 1$ and $c = 1/2$. See text for details.

former (Figs. 6(a) and (b)). This analysis therefore distinguishes the present critical point from its Ising counterpart.

To ascertain AT universality of the transition and to explore the role of the parameter $\alpha$, we obtain $\nu(\alpha)$ using finite-size scaling; the resultant plot is shown in Fig. 7(a). We find that varying $\alpha$ leads to a range of $\nu$; $\nu$ increase with decreasing $\alpha$ with $\nu \sim 3$ for $\alpha = 0$ and $\nu \to 1.24$ as $\alpha \to \infty$. This indicates that varying $\alpha$ allows one to access a section of the AT critical line; we note that in the microscopic Ashkin-Teller model $2/3 \le \nu < \infty$ on the critical line; the two limits correspond to the 4-state Potts ($\nu = 2/3$) and the Kosterliz-Thouless universality class ($\nu \to \infty$) respectively. The corresponding phase diagram (Fig. 7(b)) shows the position of this line in the $\lambda - \alpha$ plane.

Another feature of the AT critical line is that the central charge of the critical theory describing the transition remains fixed to $c = 1$ on the entire line. To further confirm that the QCPs on the line conform to AT universality, we therefore compute $S_E$ as a function of the subsystem size $l_p$ at $\lambda = \lambda_c$ and for $\alpha = 1$; the details of the computation of $S_E$ is given in App. A. A plot of $S_E$, shown in Fig. 7(c), shows a logarithmic dependence of $S_E$ as a function of $l_p$ which is consistent with the Cardy-Calabrese result [25]

$$S_E = (c/6)\ln[(2N_p/\pi)\sin(\pi l_p/N_p)]. \tag{11}$$

This allows one to extract the central charge $c$ of the conformal field theory governing the critical point (inset of Fig. 7(c)); we obtain $c = 0.895072 \pm 0.08725$ for $\alpha = 1$. Similar computations for other $\alpha$ shows that $c$ stays close to unity on the line; a plot of $c$ as a function

of $\alpha$, shown in Fig. 7(d), demonstrates this fact. We note here that since our ED can access at most $N_p = 28$, it only provides subsystems with $l_p = 4, 6, 8, 10$ and $12$. Thus the systematic error in the extraction of $c$ is higher compared to $\nu$ and $\lambda_c$; however, even with this constraint, the central charge seems to be not inconsistent, within error bars as indicated in Fig. 7(d), with AT universality [23, 24, 34]. It is clearly different from that of an Ising transition for which $c = 1/2$.

## 4.2 Landau-Ginzburg theory

The AT universality class hosts a $Z_2 \times Z_2$ symmetry. In this section, we provide a possible explanation for the origin of the additional emergent $Z_2$ symmetry at the QCP in our model. We show that this phenomenon can be understood from a Landau-Ginzburg (LG) theory for the transition. The ground state at $\lambda > \lambda_c$ is two-fold degenerate and at the QCP, a state from the excited state manifold crosses the ground state. The degeneracy ensures that if such a crossing occurs, for a periodic chain, at a momentum $k_0$ ($-\pi \leq k_0 \leq \pi$), there must be an analogous crossing at $\pi - k_0$. The critical theory, is therefore governed by a bosonic low-energy field [35, 36]

$$\Phi(x, t) = e^{ik_0 x} \varphi_1(x, t) + e^{i(\pi - k_0)x} \varphi_2(x, t) \tag{12}$$

where $\varphi_1$ and $\varphi_2$ are the low-energy fluctuations around $k_0$ and $\pi - k_0$ respectively. The invariance of the $\Phi(x, t)$ under $k_0 \to \pi - k_0$ implies that $\varphi_{1(2)} \to \varphi_{2(1)}$ under such a transformation. An effective LG theory which respects this symmetry can be written in terms of $\varphi_{1,2}(x, t)$ as

$$\mathcal{L} = \sum_{\alpha = 1,2} (|\partial_\mu \varphi_\alpha|^2 + r|\varphi_\alpha|)^2) + c_1(|\varphi_1|^4 + |\varphi_2|^4) + c_2|\varphi_1|^2|\phi_2|^2 + c_3(\varphi_1^* \varphi_2 + h.c.)^2, \tag{13}$$

where $c_1 > 0$ and we have suppressed the space time labels of $\varphi_{1,2}$ for clarity. At QCP $r = 0$ and Eq. 13 predicts condensation of both fields if either $c_2 + 2c_3 < 0$ and $c_3 < 0$ or $c_2 < 0$. In either case, $\mathcal{L}$ is two-fold degenerate; it remains invariant under $\theta_0 \to \pi - \theta_0$, where $\theta_0$ is the relative phase between $\varphi_1$ and $\varphi_2$. Its minima corresponds to $\theta_0 = 0, \pi$ for $c_3 < 0$ and $\theta_0 = \pi/2, 3\pi/2$ if $c_3 > 0$. This leads to an *additional emergent $Z_2$ symmetry*. Our analysis indicates that the microscopic $Z_2$ symmetry of the gapless phase at $\lambda > 0$, which derives its root from the subsystem symmetry of the model, necessitates a $\mathcal{L}$, with two low-energy fields; it is therefore a necessary (but not sufficient) requirement for the emergent $Z_2$ symmetry at the QCP.

## 5 Mapping to a 1D spin model

In this section, we present the mapping of $H$ model to a modifed PXP Hamiltonian [30–32, 36, 37]. It is well known that PXP models provide a description of the low-energy physics of ulrtacold Rydberg atoms [26–28, 30–32, 37, 38]; we show that our model reduces to the PXP model for $\alpha \to \infty$ thus allowing for possible experimental relevance of the phase transition studied in the original boson model.

To this end, we first map a state with two bosons on the transverse diagonal $d_t$ of a plaquette to a up-spin at its center as shown in Fig. 8(a): $|t_j\rangle \to |\uparrow_{x_j}\rangle$, where $x_j$ denotes the coordinate of the center of the plaquette. Similarly, a state with two bosons on the longitudinal diagonal $d_\ell$ of a plaquette is mapped to a down-spin: $|\ell_j\rangle \to |\downarrow_{x_j}\rangle$ (see Fig. 8(b)). Crucially, the presence of subsystem symmetry does not allow states with two neighboring plaquettes each having two bosons on $d_t$; such a state correspond to $n_0 = 2$ and are dynamically disconnected from states with $n_0 = 1$. This allows one to realize the PXP constraint of

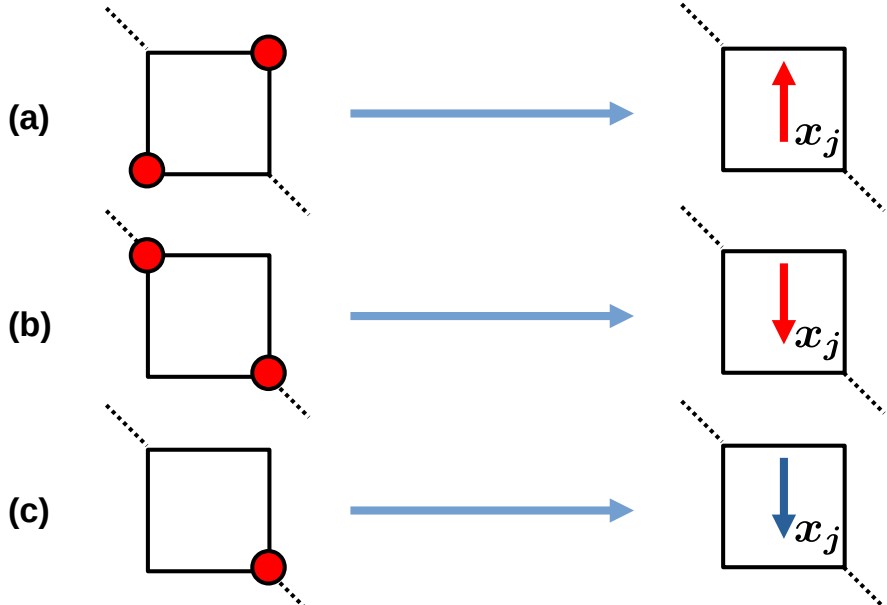

Figure 8: Schematic representation of mapping of bosons to PXP spins. (a) A boson state with two bosons on the transverse diagonal is mapped to an up spin at the center of the plaquette with coordinate $x_j$. (b) A boson state with two bosons on the longitudinal diagonal is mapped to a flippable down spin. (c) A lone boson is mapped to an unflippable down spin. See text for details.

not having two neighboring up-spins. Finally, as shown in Fig. 8(c), we denote a lone boson on in a given plaquette as an unflippable down spin since the action of $H_0$, in contrast to the standard down-spins defined earlier, can not flip it.

To express the boson Hamiltonian in terms of these spins we define the following operators

$$\tau^z_{x_j} = |t_j\rangle\langle t_j| - |\ell_j\rangle\langle \ell_j|, \quad \tau^x_{x_j} = |t_j\rangle\langle \ell_j| + |\ell_j\rangle\langle t_j|. \tag{14}$$

We also define a projection operators $P_{x_j} = (1 - \tau^z_{x_j})/2$ which projects to $|\downarrow_{x_j}\rangle$. Noting that a flip of spin from down to up is possible only if the neighboring sites also have down spins (either flippable or unflippable), we can write

$$H_0 = -w \sum_{x_j} \tilde{\tau}^x_{x_j}, \quad \tilde{\tau}^x_{x_j} = P_{x_j-1} \tau^x_{x_j} P_{x_j+1} \tag{15}$$

To write $H_1$ in terms of spin operators, we note that $H_1$ does not count unflippable spins. Thus we need to count only flippable down spins. It is easy to see from the boson states such spins are the ones which are flanked by two other down spins on neighboring sites. These considerations lead one to the spin Hamiltonian

$$H_1 = -\lambda \sum_{x_j} \left( \alpha \frac{1 + \tau^z_{x_j}}{2} - P_{x_j-1} \frac{1 - \tau^z_{x_j}}{2} P_{x_j+1} \right)$$

$$= -\lambda \sum_{x_j} (\alpha \tau^z_{x_j}/2 - P_{x_j-1} P_{x_j} P_{x_j+1}), \tag{16}$$

where in the second line, we have ignored an irrelevant constant. We note that the last term in Eq. 16 which arises due to the presence of unflippable down spins (or lone bosons in a plaquette) distinguishes the model from the PXP model in a magnetic field [36].

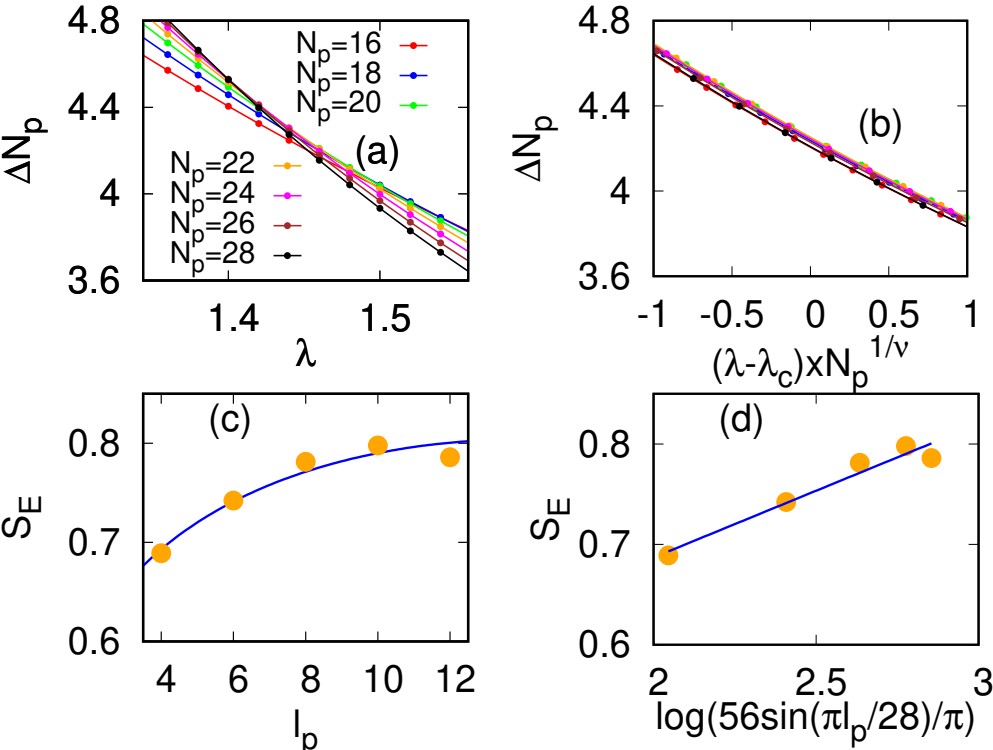

Figure 9: Finite-size scaling analysis for $\alpha \to \infty$ starting from $H'$ (Eq. 17). (a) Plot of $\Delta N_p$ as a function of $\lambda$; the curves cross at $\lambda = \lambda_c \simeq 1.45097 \pm 0.002631$ indicating $z = 1$. (b) Plot of $\Delta N_p$ as a function of $N^{-1/\nu}(\lambda - \lambda_c)$ leading to optimal scaling collapse for $\nu \simeq 1.24173 \pm 0.02461$. (c) Plot of $S_E$ as a function of $l_p$ showing logarithmic dependence of $S_E$ on $l_p$ (d) Plot of $S_E$ as a function of $\ln[(2N_p/\pi)\sin(\pi l_p/N_p)]$ leading to $0.797501 \pm 0.1052$. For all plots $w = 1$. See text for details.

Next, we study the limit $\alpha \to \infty$ with $\lambda \alpha / w$ being held fixed and kept finite. In this limit, one can ignore the last term in Eq. 16 and map the plaquette boson model to the PXP model in a magnetic field. This yields a spin Hamiltonian $H'$ given by

$$H' = -\sum_j (w\tilde{\tau}^x_{x_j} + \lambda \tau^z_{x_j}) \tag{17}$$

where $\lambda \to \lambda \alpha / 2$. Such a chain with periodic boundary condition (PBC) has been studied in detail [36,37] and is known to have Ising critical point at $\lambda_c/w \simeq 1.31$. However, the criticality in the model with open boundary condition (OBC) turns out to be different. One of the source of this difference stems from the nature of the ground state at large $\lambda/w$. The PXP chain with PBC has a gapped ground state at large $\lambda/w$; in contrast, in the presence of OBC, the ground state at large $\lambda/w$ is gapless. Such a gapless states occurs due to superposition of all Fock states with alternate up and down spins along with a single domain wall. Note that the presence of such a domain wall in the ground state sector can be traced back to the presence of subsystem symmetry of the original boson Hamiltonian, which, for a given $N = N_p + 1$, necessitates the presence of a loan boson in the ground state manifold. The energy band of these excited states has a width $\sim 1/\alpha$ as can be deduced from Eq. 7.

A finite-sized scaling analysis starting from $H'$ confirms the difference in criticality of the PXP chain with OBC from the one with PBC. This is shown in Fig. 9. Our analysis indicates a strong finite size effect for a chain with OBC at large $\alpha$; this feature is to be distinguished from the chain with PBC and may arise from the difference in nature of the ground states in the two cases at large $\lambda/w$. The numerical results indicate $\lambda_c/w = 1.45097 \pm 0.002631$, $\nu = 1.24173 \pm 0.02461$ and $c = 0.797501 \pm 0.1052$ which is quite different from Ising universality; these exponents are however, consistent with AT universality after allowing for strong finite-size corrections for estimation of $c$ at large $\alpha$.

Before ending this section, we note that it is known for an Ising chain that a change in boundary condition may change $c$ [39]; we numerically find for our model that this is indeed the case for $\alpha \to \infty$. At finite $\alpha$, the third term provides an additional three-site interaction which changes $\nu$; such a multi-spin term therefore provides a way to access the AT line in these experimentally realizable Rydberg systems [26–29]. Engineering such terms in Rydberg arrays requires further work which we leave for future studies.

# 6 Discussion

In this work, we have studied a boson model with subsystem symmetry and have shown that the presence of such subsystem symmetry can change the universality of the quantum phase transition of the model. Our analysis indicates that the model has two phases; one of them occurs at large negative $\lambda$ and is gapped. The second phase, occurring at at large positive $\lambda$ is gapless in the thermodynamic limit; moreover, it breaks a discrete $Z_2$ symmetry. Our analysis unravels the key role played by the subsystem symmetry behind the presence of the latter phase.

These two phases of the model, as expected, are separated by a quantum critical point. However, in contrast to the standard expectation, we find that the universality class of this transition is not Ising; instead, it belongs to the AT universality class. The parameter $\alpha$ of the model (Eq. 2) can be tuned to access the AT critical line with variable $\nu$ but constant $c$. Our analysis indicates that the boson model can access a part of the AT critical line ($3 \leq \nu \leq 1.24$). The computed central charge shows strong fluctuation; this is due to the fact that using ED, one can access $N_P \leq 28$ allowing subsystem sizes $4 \leq \ell_p \leq 12$. However, even with this constraint, the values of $c$ obtained seem to be very different from its value for Ising criticality ($c = 1/2$).

Moreover, the tunability of $\nu$ along with a value of $c \neq 1/2$ certainly seems to suggest that for the system sizes accessible by ED, the universality class of the transition is not inconsistent with Ashkin-Teller. However accessing larger system sizes is necessary for a more definitive conclusion. In this respect we note that while the presence of the constraint does not allow a straightforward numerical analysis of this model using density-matrix-renormalization group (DMRG) or Quantum-Monte-Carlo (QMC) methods, certain recent methods using sweeping cluser QMC algorithm [40] or DMRG of constrained dimer model [41] may prove useful. These endeavors constitute future directions of research.

We also provide an exact mapping of the boson model to the PXP spin model in a magnetic field with open boundary condition describing a ultracold Rydberg atom chain. This leads to the possibility of its realization using ultracold atom platforms. The two models are identical in the limit $\alpha \to \infty$ with $\lambda \alpha / w$ held fixed and finite. For finite $\alpha$, we find that the boson model is equivalent to a PXP model in a magnetic field together with an additional three-spin term. The variation of the strength of this three-spin term (which is same as varying $\alpha$ keeping $\lambda / w$ fixed) allows one to access the AT critical line. The realization of such a three spin term in experimental Rydberg platforms may be of future interest.

The Hilbert space dimensions (HSD) of $H_s$ and hence $H$ scales as $\zeta^L$; $H_s$ therefore provides easier numerical access to the physics of the AT critical line compared to a standard spin model with HSD $4^L$ [42, 43]. This may be useful for studying non-equilibrium quantum dynamics of the model across the AT critical line; this might be another subject for future study [44, 45].

In conclusion, we have demonstrated the emergence of AT criticality in a constrained bosonic model with subsystem symmetry. We point out the crucial role of subsystem symmetries in determining the universality class of the critical theory. Our analysis also shows that the model is closely related to the well-known PXP model relevant for Rydberg atom arrays and forms the most convenient platform for numerical studies of the AT critical line.

*Acknowledgements*: KS thanks DST, India for support through SERB project JCB / 2021 / 000030 and S. Sachdev for discussion. AM thanks DST for support through SERB NPDF grant PDF / 2022 / 000124.

## A    Entanglement entropy computation

In this section, we present details of calculations for the entanglement entropy $S_E$. We start with the effective spin Hamiltonian given by Eq. 16 of the main text. We partition the spin chain of length $L$ into subsystems $A$ of length $\ell_A$ and $B$ of length $\ell_B$. The ground state of the system is then represented by

$$|\psi\rangle = \sum_{i,\mu} c_{i,\mu} |i, \mu\rangle \tag{18}$$

where the labels $i$ and $\mu$ correspond to subsystems $A$ and $B$ respectively. As discussed in Refs. [44, 45], one needs to carry out this partitioning in a manner consistent with the constraint condition. The density matrix for the ground state can then be written as

$$
\begin{aligned}
\rho &= |\psi\rangle \otimes \langle\psi| \\
&= \sum_{ij}\sum_{\mu\nu} c_{i\mu}^* c_{j\nu} |[i][\mu]\rangle \otimes \langle[j][\nu]|,
\end{aligned}
\tag{19}
$$

where $[i], [j]$ represent the set of indices associated with subsystem $A$ and $[\mu]$ and $[\nu]$ with subsystem $B$. The $c_{i\mu}$'s are the components of the ground state wavefunction in the Fock basis.

To compute the entanglement entropy, we need to trace over the

$$\rho_A \;=\; Tr_B(\rho) = \sum_\sigma \sum_{ij} \sum_{\mu\nu} c^*_{i\mu} c_{j\nu} \delta_{\mu\sigma} \delta_{\nu\sigma} \,|[i]\rangle \otimes \langle[j]|$$

$$\implies \rho_A^{ij} \;=\; \sum_\sigma c^*_{i\sigma} c_{j\sigma}. \tag{20}$$

The von-Neuman entanglement entropy is defined in terms of $\rho_A$ as

$$S_E = -tr[\rho_A \ln \rho_A]. \tag{21}$$

Denoting the eigenvalues of $rho_A$ are $\{\lambda_i\}$, one can compute $S_E = -\sum_i \lambda_i \ln \lambda_i$. In this work, we have used this formalism to compute the entanglement entropy $S_E(\ell_A)$ for which $\ell_A = L - \ell_b = \ell$.

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
