# Peer review of "Emergent Ashkin-Teller criticality in a constrained boson model"

_SciPost Physics_

## Round 3 · Referee Report · Anonymous (Referee 1) · 2024-10-14

Report
The authors proposed a constrained boson model on a string of boson arrays, which has to distinct states: a Z2 gapless state and a Z2 symmetry breaking gapped state. The existence of the Z2 gapless phase is demonstrated by a degenerate perturbation theory and finite-size exact diagonalization up to N=29 bosons. They argued the subsystem symmetry, in this case, the conservation of particle number on the vertical/horizontal links change the universality class of the quantum phase transition from a Ising universality class to a Ashkin-Teller type transition with an addition emergent Z2 symmetry, making the symmetry group Z2xZ2. The authors support this with the scaling of the excited states and compare with those of the transverse Ising mode, varying critical exponent \nu with \alpha and the central charge stays c=1.
Physics in systems under local constraints actively research subjects in recent years and this paper is timely. However, there are several things I hope the authors can clarify. 1. To show that the transition is AT type, it is crucial to show that there exists three gapless modes in the thermodynamic limit. However, Fig.6(b) not only shows crossing at largest two or three sizes and the crossing point \lambda/w has a significant shift away from the 0.7 as established in the lower excitations. It is hard to believe this should scales to the same critical point in the thermodynamic limit. This also applies to the entanglement entropy and central charge calculations. Although this transition is definitely not Ising, to argue the transition is AT needs stronger numerical evidences.
- The authors argue that due to the constraints, this model can not be studied efficiently using QMC or DMRG. Since the model shows similarity to a quantum dimer model, where some efficient algorithms such as the so-called sweeping cluster algorithm have been proposed for efficient simulations. I wonder if one can add some terms in the model to enforce the dimer constraint?
Recommendation
Ask for minor revision
We thank the referee for a careful reading of the manuscript. Our response to their comments are as follows:
- To show that the transition is AT type, it is crucial to show that there exists three gapless modes in the thermodynamic limit. However, Fig.6(b) not only shows crossing at largest two or three sizes and the crossing point \lambda/w has a significant shift away from the 0.7 as established in the lower excitations. It is hard to believe this should scales to the same critical point in the thermodynamic limit. This also applies to the entanglement entropy and central charge calculations. Although this transition is definitely not Ising, to argue the transition is AT needs stronger numerical evidences.
We agree with the referee that we can make a more concrete statement only after studying larger systems which is currently beyond the scope of the ED studies that we perform. At the length scale which we could access, we could certainly perceive the difference in crossing of our model with the Ising; this is seen by comparing Figs 6a and 6d. This crossing corresponds to the second gapless mode in our model (although its position shifted due to finite size effect) and is clearly distinct from the corresponding behavior of the Ising model; however, we agree with the referee that the crossing corresponding to \Delta_{30} (Fig 6b) requires access to larger system size. We have now added a discussion of this point in the discussion section.
Regarding identification of the universality of this transition with Ashkin-Teller, we note that the variation of \nu with the parameter \alpha showing weak universality and a central charge value closer to unity that to 1/2 is indicative of this. Also the analytical Landau-Ginzburg arguments seems to point to an emergent Z_2 \times Z_2 symmetry at criticality. We of course agree with the referee that one needs to access higher system size for a more definitive statement. We have now pointed this out in the discussion section of the manuscript.
- The authors argue that due to the constraints, this model can not be studied efficiently using QMC or DMRG. Since the model shows similarity to a quantum dimer model, where some efficient algorithms such as the so-called sweeping cluster algorithm have been proposed for efficient simulations. I wonder if one can add some terms in the model to enforce the dimer constraint?
We thank the referee for pointing out this aspect. Indeed our model can be mapped onto dimer model on a two leg ladder ( with the centers of the plaquettes of the ladder acting as the sites of the equivalent PXP chain) but with the unconventional relative sign of the on-site dimer term ( which has not been studied earlier, to the best of our knowledge). It is possible that this can be studied by sweeping cluster QMC algorithm (PRB 99, 165315 (2019)) . A similar implementation may probably be done by DMRG following the algorithm by Mila et al ( Scipost Phys 6 33 (2019). We have now added a discussion along with the refs regarding these possibilities in the main text. We keep these problems as subjects of future studies.
We hope that the above changes will be acceptable to the referee.

Anonymous on 2024-09-12 [id 4768]
I apologize for not being able to review this paper.
I went through it and my main concern is that the claimed Z2 phase is in fact Z4. Selected boundary conditions might prevent the authors from seeing the true bulk degeneracy, in particular, on such small clusters. Said this, the appearance of the Ashkin-Teller transition is therefore not surprising. Furthermore, as the authors do not include longer-range interactions, it is also natural that the transition remains conformal.
Anonymous on 2024-11-02 [id 4928]
(in reply to Anonymous Comment on 2024-09-12 [id 4768])Response: We thank the referee for the comment. However, we disagree with it.
In our work we have studied both periodic and open boundary condition and found a microscopic
Z_2 symmetry in both cases. This symmetry corresponds to reflection through the center of the chain for open boundary condition and a k to \pi-k symmetry for periodic boundary condition as discussed in details in the paper. We do not see occurrence of a microscopic Z_4; since the comment does not mention the source of this Z_4 either, this claim remains cryptic and we disagree with it.

---

## Editorial Decision

unknown